# Morula Kernel Cake (*Sclerocarya birrea*) as a Protein Source in Diets of Finishing Tswana Lambs: Effects on Nutrient Digestibility, Growth, Meat Quality, and Gross Margin

**DOI:** 10.3390/ani13081387

**Published:** 2023-04-18

**Authors:** Leonard Baleseng, Othusitse Madibela, Christopher Tsopito, Molebeledi Mareko, Wame Boitumelo, Moagi Letso

**Affiliations:** Department of Animal Sciences, Faculty of Animal and Veterinary Sciences, Botswana University of Agriculture and Natural Resources, Private Bag 0027, Gaborone, Botswana

**Keywords:** morula kernel cake, nitrogen retention, organoleptic attributes, lamb

## Abstract

**Simple Summary:**

The inclusion of morula kernel cake (MKC) as a source of protein in finishing diets of Tswana lambs positively supported growth and enhanced sensory attributes of cooked meat steaks. Feed intake, average daily gain (ADG), feed conversion ratio (FCR), and carcass traits were similar for lambs fed on diets containing Lucerne (commercial diet), sunflower seedcake, or MKC. However, lambs from a commercial diet (CD) had a better nitrogen retention than lambs from a sunflower seedcake diet (SCD). Nitrogen retention in lambs fed a morula kernel cake diet (MKCD) was numerically better than those from SCD. The nutritional meat quality parameters were similar across the dietary treatments and were within the recommended reference values. Additionally, meat steaks of lambs that consumed MKCD were rich in oleic acid while meat steaks of lambs that ate SCD had no oleic acid. Furthermore, the gross margins from lambs fed either MKCD or SCD were better than those fed CD. This was attributed to lower variable costs of feed that accounted to 51% of production costs in lambs fed either MKCD or SCD when compared to lambs fed CD, which had variable costs of 71% apportioned to feed.

**Abstract:**

A trial evaluated growth performance, carcass characteristics, meat quality, and economic returns of fattened lambs fed on diets containing different protein sources. Six castrated male Tswana lambs per treatment were used in a completely randomised design (CRD) trial and fed on complete diets containing Lucerne (CD; commercial diet), morula kernel cake (MKC), or sunflower seedcake (SC) as protein sources over a 103-day experiment. No significant differences (*p* > 0.05) were observed in the dry matter intake, final body weight, average daily gain, and FCR. This was attributable to all the diets providing an equal supply of nutrients to the lambs. Meat quality attributes and proximate composition values were similar (*p* > 0.05) across the treatments. *Longissimus dorsi* muscle organoleptic quality did not differ (*p* > 0.05) across the treatments. The gross margin analysis was significantly greater (*p* < 0.05) when feeding SCD than feeding CD and was intermediate for lambs fed MKCD. Morula kernel cake (*Sclerocarya birrea*) can be used for fattening lambs when common protein sources are either not available or expensive.

## 1. Introduction

Protein and energy ingredients contribute to a higher cost of livestock production [1]. The major conventional plant protein sources used globally in livestock feeding are soybean meal, canola cake, cottonseed meal, and sunflower cake [2]. The demands for protein sources is very high, especially that protein is one of the critical nutrients needed for growth by young animals [1]. At the same time, there is currently a high demand for animal protein food coupled with an increase in the human population [3]. Furthermore, consumers nowadays are increasingly demanding safe and quality food from animal products, especially meat. Meat is a good source of protein, fat, vitamin B, and minerals in human diets [4]. However, there has been a concern about saturated fatty acids from meat that are purported to predispose cardiovascular diseases [4] in humans. According to Chiofalo et al. [5] rumen microbes hydrogenate dietary unsaturated fatty acids resulting in the production of saturated fatty acids that are absorbed and deposited in animal tissues. The deposition of fat in tissues is determined by the animal’s diet and sex [4]. Additionally, Junkuszew et al. [6] stated that dietitians recommend consumption of meat low in fat and rich in n-3 and n-6 fatty acids to avoid heart-related diseases in humans. Therefore, this calls for ruminant nutritionists to look for novel feed ingredients that will promote the accumulation of healthy fat in animal tissues and to meet holistic market requirements for finished animals.

The main challenge faced by livestock farmers to produce a consistent supply of quality animal products is the inadequate supply of required nutrients due to seasonal fluctuation of forages from natural pastures. As a result, some farmers, as stated by Yagoubi et al. [7], end up feeding their animals with low quality forages or allowing them to graze on degraded natural pastures, especially during the dry season. Obviously, such animals will grow slowly and take a longer period to reach market weights. Therefore, provision of protein or energy supplements will promote better utilisation of poor quality roughages by ruminants and synthesis of microbial proteins. Proteins that reach the small intestine of ruminants are made up of digestible microbial proteins and digestible rumen undegradable proteins, which in summation are regarded as metabolizable proteins [8]. According to Ruzic-Muslic et al. [9], microbial protein is not adequate to supply the amino acids required for optimum animal growth; therefore, protein sources with high rumen undegradable protein (RUP) are required for better growth performance. Most conventional plant protein sources used to supplement protein and energy in ruminants are very expensive for resource limited farmers. Unconventional oilseed cakes like MKC can be explored further for use in ruminant diets to lower the cost of livestock production. Morula kernel cake is a residue remaining after oil extraction from the morula nut [10]. Malebana et al. [11] have shown that MKC is a good source of protein, fibre, fatty acids, and minerals. Earlier researchers demonstrated that MKC can be used in diets of goats [10] and dairy cattle [12]. However, information on its effects on meat quality is lacking. Consumer decision on meat quality is based on meat palatability components such as flavour and tenderness [13,14]. This study was designed to assess the effects of an MKC-based diet on feed intake, weight gain, carcass characteristics, meat quality, sensory traits, and gross margin (GM) analysis when compared to a sunflower seedcake-based diet or a commercial diet.

## 2. Materials and Methods

### 2.1. Ethics Statement

The care of animals was in accordance with the guidelines of Council for the International Organisation of Medical Sciences [15] and the study was approved by Botswana University of Agriculture and Natural Resources (BUAN) animal ethics committee (approval number BUAN-AEC-2019-01).

### 2.2. Animals, Facilities, and Experimental Procedure

Eighteen castrated male Tswana lambs aged eight months on average were obtained from BUAN farm. These lambs were each weighed for two consecutive days to determine their initial weights. Each lamb was injected subcutaneously with 1 mL of Ivermax (Copal Grimed Pty, Ltd., Pretoria; South Africa) before the experiment began to treat for internal parasites such as round worms and tapeworms. These lambs were divided into three groups balanced for weight (16.9 ± 0.7 kg), and the groups were randomly distributed accordingly to the three dietary treatments of a commercial diet (CD), morula kernel cake diet (MKCD), and sunflower seedcake diet (SCD). There were six animals per treatment and each animal was randomly allocated to a pen (2.5 m × 1 m) in a completely randomized design (CRD). The concrete floors of pens were cleaned daily before providing fresh feed and water.

### 2.3. Diets and Feeding

Two treatment diets were formulated using Aries software (Davis, CA, USA) and prepared using MKC and SC as protein sources to meet the nutrient requirements (protein = 14.7% and energy = 78% of total digestible nutrients (TDN)) of lambs aged approximately eight months old, as outlined by National Research Council [16]. Morula kernel cake was purchased from DLG Naturals (Gabane, Botswana) while sunflower seedcake was purchased at Arona Pty Ltd. (Gaborone, Botswana). A purchased commercial diet (CD) for lambs with Lucerne (*Medicago sativa*) as a protein source was used as the control diet. Other ingredients in the CD were yellow maize, maize bran, feed lime, ammonium chloride, liquid molasses, salt, and vitamin premix (feed dealer; personal communication). The ingredients for MKCD and SCD were prepared as shown in Table 1. The lambs were adapted to the diets and pens for 7 days. Allocation of the experimental diets during the adaptation period was restricted to avoid digestive disorders. After the adaptation period, each lamb was fed *ad libitum* daily with 15% allowance of leftovers in their individual pens for 103 days. Feed leftovers and feed given were weighed daily using a platform scale (Digital scale; model DS-530, Teraoko Seiko, Tokyo, Japan).

### 2.4. Digestibility Experiment

After 90 days of the feeding experiment, four lambs from each treatment were randomly selected and transferred to metabolic crates for a digestibility experiment. The same treatments from the growth experiment were used accordingly. The lambs were kept individually in a metabolism crate designed for the collection of faces and urine in separate containers. The lambs were adjusted to metabolic crates and faecal bags for 7 days. Thereafter, a period of 6 days was used for data and sample collection. Fresh water was provided daily for each lamb *ad libitum*. Each morning, before providing fresh feed, the leftovers from the previous day were collected and weighed to determine daily feed intake. Daily samples of feed were collected and composited per treatment for chemical analysis.

Daily total faces for each animal were collected and weighed in the morning before feeding and providing water. Approximately 10% of the faecal material was collected, oven dried at 70 °C for 72 h, and bulked for each animal to use for chemical analysis [17]. The urine of each animal was collected in a plastic container containing 25 mL of 10% sulphuric acid (H_2_SO_4_) [17]. Thereafter, 10% of the measured daily urine was sampled and a composite sample for each animal was poured in a labelled screw-capped plastic bottle and stored in the refrigerator at 4 °C pending analysis of nitrogen. Apparent digestibility (AD), nitrogen balance (NB), basal endogenous nitrogen (*BEN*), and nitrogen retention (NR) were calculated as described below: Nitrogen balance (NB) = Nitrogen Consumed-Nitrogen in excreta (faces + urine). Basal endogenous nitrogen (BEN) was calculated using the following equation by McDonald et al. [18]: $BEN(g/d)=\left(0.35+0.018\right)\times BW^{0.75}$, *BW* stands for body weight. The corrected value for Nitrogen retention is: $NR\left(g/d\right)=NB-BEN$. Apparent digestibility (AD) of nutrients was calculated using the following formula [18,19]:$$AD=\frac{Nutrient\;consumed\;-\;Nutrient\;in\;faeces}{Nutrient\;consumed}\;\times \;100$$

### 2.5. Slaughter Procedure

After 103 days of feeding, all the lambs were weighed before morning feeding for 2 consecutive days to determine final weight using a mobile walk-in scale (Crane scale: Tal tec Pty, Ltd., Johannesburg, South Africa). Thereafter, the lambs were humanely slaughtered at a commercial abattoir in the morning after a night fast. The lambs were electrically stunned and immediately bled by cutting the jugular vein and carotid arteries using a sharp knife. The carcass was then weighed to obtain the hot carcass weight (HCW) using a spring balance. The empty body weight (EBW) was obtained by subtracting the rumen and intestinal contents weight from final weight [8]. After carcass weighing, the *Longissimus dorsi* muscle was sampled between the 5th and 13th rib from the left and right side of each carcass for use in sensory analysis and meat quality determination, respectively.

### 2.6. Chemical and Technological Analysis

Samples of feed, faecal material, and urine collected during the trial were analysed in duplicate. Both feed and faecal samples were analysed for dry matter (DM), organic matter (OM), ash, and ether extract (EE) following procedures of A.O.A.C. [20] and crude protein (CP) according to A.O.A.C. [21]. Neutral detergent fibre (NDF), acid detergent fibre (ADF), and acid detergent lignin (ADL) were analysed sequentially using Ankom^220^ Fibre Analyser (New York, NY, USA), as described in the Ankom^220/220^ Fibre Analyser manual [22]. Gross energy was determined using CAL3K-S Oxygen bomb calorimeter system (Digital data system, Randburg, South Africa) following the guidelines from manual of DDS Calorimeters [23]. Urine samples were analysed for nitrogen following procedures of A.O.A.C. [21]. The chemical composition of experimental diets is reported in Table 2.

Chemical analysis of meat was done using six replications per treatment. Meat samples of the *Longissimus dorsi* muscle were sent for proximate analysis (moisture, crude protein, fat, and ash) to the National Food Technology Research Centre (NFTRC), Kanye, Botswana. Fatty acid analysis in meat was determined following procedures of O’Fallon et al. [24] to produce fatty acid methyl esters (FAME). Fatty acids were profiled at University of Botswana (UB) using gas chromatography (temperature ranged from 200 °C to 270 °C, GC syringe size was 1 µL and automated, capillary tube size was 30 m × 0.25 m × 0.25 µm; Agilent Technologies, 7890A, GC system, Santa Clara, CA, USA), mass spectrometry (MSD, GC, Agilent Technologies, 5975C, Santa Clara, CA, USA), and a personal computer for quantification. The *Longissimus dorsi* muscle technological analysis entailed pH, colour, tenderness, and meat sensory analysis. Meat pH_24_ was measured using a pH meter (Model HI99163, USA) and meat colour was determined after 24 h of storage using precision meat colorimeter (Miniscan, model NR20XE, 3NH Technology Co, Ltd., Shenzhen, China). The instrument was calibrated using both white tile and a black tile before measurements were taken, while tenderness of cooked meat steaks was determined using Digital firmness tester (Model Agro R15, Serqueux, France), as described by Machete et al. [14]. Sensory evaluation of lamb steaks was done using untrained taste panel of 22 persons of mixed gender aged 26 to 55 years. The panellists were inducted on how to evaluate the meat samples before carrying out the exercise. The taste panel evaluated cooked meat steaks on the following: appearance, taste, flavour, juiciness, and overall impression. The rating scale used ranged from 1 = dislike extremely to 9 = like extremely, as described by Morlein [25]. Composite meat samples of 1.1 kg per treatment were cooked separately. The meat samples were diced into approximately 1 cm^3^ and put into stainless pots. Meat was cooked on a medium heat using an electric stove. Salt was added after 15 min of cooking and the duration of cooking was 1 h and 20 minutes. Thereafter, cooked meat samples were placed in coded plates after cooling for individual sensory evaluation. After evaluation of meat steaks from each treatment, the panellists were requested to sip water to neutralise their senses.

### 2.7. Gross Margin Analysis

The amount of feed consumed by each lamb over a period of 103 days of feeding was quantified and feed costs for each treatment were calculated. The variable costs included labour, drugs, feeds, abattoir costs, transportation of animals to abattoir, and purchase cost of weaner lamb. The value of the carcass and edible offals for each treatment were summed up to estimate revenue for each treatment. Gross income was calculated from the sale of each animal carcass and edible offals. Therefore, Gross Margin (GM) = Gross Income (GI)—Variable Costs (VC).

### 2.8. Statistical Analysis

Data on animal performance, meat quality, sensory evaluation, and gross margin analysis were analysed as CRD using the general linear model (GLM) procedure of SAS [26], and are represented by $Y_{ijk}=\mu +T_{i}+\varepsilon_{ij}$ where *Y_ijk_* = is the *K^th^* observation of the *i^th^* treatment, *µ* = is the overall mean, *T_i_* = is the fixed effect of the *i^th^* treatment (*i* = 1, 2, 3), and $\varepsilon_{ij\;}$ = random residual error. The significance of the treatment effects was verified using F-test and means were separated using Tukey’s procedure. Level of significance was declared at *p* < 0.05, and for tendencies, significance was declared at *p* < 0.1.

## 3. Results

### 3.1. Nutrient Intake

The measured nutrient intake was similar (*p* > 0.05) across the dietary treatments except for ether extract intake and acid detergent fibre (ADF) intake (Table 3). The ether extract intake was higher (*p* < 0.05) in lambs consuming MKCD than for lambs fed CD or SCD. On the other hand, ADF intake was significantly (*p* < 0.05) lower in lambs fed CD than in lambs that consumed MKCD or SCD.

### 3.2. Apparent Nutrient Digestibility

The digestibility of CP was higher (*p* < 0.05) in lambs that ate CD but was lower and similar in both lambs consuming MKCD or SCD (Table 4). There was a tendency for treatment effect in digestibility of dry matter (*p* = 0.08) and organic matter (*p* = 0.07). However, digestibility of other measured parameters was the same across the treatments.

### 3.3. Nitrogen Balance

Parameters for the nitrogen balance trial are shown in Table 5. Lambs fed CD had a significantly higher (*p* < 0.05) nitrogen retention and retention-intake percentage, while lambs from SCD had the lowest nitrogen retention and retention-intake percentage. Faecal-N (*p* = 0.06) and retention-A (*p* = 0.07) tended to be different between treatments. Other measured parameters were similar (*p* > 0.05) across the treatments.

### 3.4. Growth Performance

Growth performance and carcass traits are shown in Table 6. The were no differences (*p* > 0.05) observed in the measured parameters between treatments.

### 3.5. Physico-Chemical Attributes and Proximate Composition of the Longissimus Dorsi Muscle from Lambs

The physico-chemical attributes and proximate composition of the *Longissimus dorsi* muscle are shown in Table 7. The diets did not affect (*p* > 0.05) the measured parameters.

### 3.6. Fatty Acid Profile of Longissimus Dorsi Muscle

The content of fatty acids is presented in Table 8. The *Longissimus dorsi* muscle from lambs fed SCD had a significantly higher (*p* < 0.05) palmitic acid level than the *Longissimus dorsi* muscle from lambs fed CD or MKCD. Oleic acid content was significantly higher (*p* < 0.05) in *Longissimus dorsi* muscles from lambs fed MKCD and was not detected in *Longissimus dorsi* muscles from lambs fed on SCD. Monounsaturated fatty acids were higher (*p* < 0.05) in *Longissimus dorsi* muscles from lambs fed on MKCD and lower in *Longissimus dorsi* muscles from lambs fed SCD. On the other hand, polyunsaturated fatty acids were significantly (*p* < 0.05) higher in *Longissimus dorsi* muscles from lambs fed CD than from *Longissimus dorsi* muscles from lambs fed MKCD or SCD. There was tendency (*p* = 0.06) for higher saturated fatty acids in lambs fed SCD.

### 3.7. Sensory Analysis

Regarding the meat sensory attributes of cooked meat steaks, no differences (*p* > 0.05) between treatments were observed for all the measured parameters (Table 9). However, cooked meat steaks from lambs that consumed MKCD had a tendency (*p* = 0.09) to rank high on overall impression.

### 3.8. Gross Margin Analysis

The gross margin analysis of treatment diets is shown in Table 10. The variable costs were significantly (*p* < 0.05) higher from lambs fed CD than from lambs fed other treatments (MKCD and SCD). Additionally, lambs fed CD had the lowest (*p* < 0.05) gross margin value among treatment diets.

## 4. Discussion

### 4.1. Nutrient Intake and Digestibility

Nutrient intake is a function of digestibility of feed ingested. Diets with high digestibility have high intake [18] of nutrients. According to Zahao et al. [27], efficient livestock production is dependent on precise knowledge of their energy and nutrient requirements. Dry matter intake (g/day) in the current study was similar across the treatments and total dry matter intake as a percent body weight was within the recommended range of 2 to 6% for sheep [28]. Similar dry matter intake across the treatments is attributable to similar intake of CP across the treatments (Table 3). This suggests that nitrogen supplied by the diets in the rumen resulted in an increased microbial population that digested the ingested feed thereby culminating in the current observed DMI. Ahmed et al. [29] reported non-significant DMI by desert rams fed on complete diets with different nitrogen sources (cotton seed cake, sesame seed cake, groundnut cake, and sunflower seedcake). Since DMI was similar across treatments and was associated with CP intake, any difference in animal response will be due to a difference in protein quality and the digestibility of other nutrients. The amount of ether extract (EE) in MKCD was higher at 9.4% (Table 2), with 84 g/d EE intake (Table 3). However, the nutrient intake (crude protein, neutral detergent fibre, and acid detergent fibre) was not depressed due to this high EE intake. Nevertheless, it has been noted by Fiorentini et al. [30] that different sources of lipids with different fatty acid profiles affect intake and digestibility differently. The present study suggests that rumen microbes, which digest fibre, were not impeded by the amount of EE in MKCD. In contrast, Mlambo et al. [10] reported that supplementary feeding of MKC in the diets of Nguni goats significantly reduced the total organic matter intake. Disparities observed between the present study and that of Mlambo et al. [10] could be attributed to genotype of animals, different diet composition, and trial management. In addition, despite similarities in CP intake across the treatments in the present study, lambs fed CD had a higher digestibility of CP (92.3%), while those offered MKCD and SCD had a lower digestibility of CP. This may be explained by the higher intake of ADF or possible presence of anti-nutritional factors from MKC and sunflower seedcake. According to Kamanula et al. [31], MKC has tannins and sunflower seedcake contains chlorogenic acid [32], both of which can bind enzymes that digest protein. This present study did not evaluate phytochemical properties.

### 4.2. Nitrogen Balance (NB)

Evaluation of NB is an important and accurate tool for appraising the value of protein sources for ruminants [33]. Different intake of nitrogen in the present study did not show a statistically significant difference. This is attributed to similar protein content and supply by the dietary treatments. The fact that nitrogen digestibility and NB were lower in SCD than in CD suggests that nitrogen from the Lucerne was readily available to rumen microbes and RUP from the diet was used by the host animal for tissue development and other bodily functions. Nitrogen digestibility and NB of lambs fed MKCD and SCD were similar. However, sheep fed MKCD had a higher but non-significant nitrogen retention (6.4 g/day compared to 3.9 g/day for SCD animals). This can be explained by MKC providing RUP of 11.2% to the diet [34] while sunflower seedcake provided a RUP of 9.8% to the diet [35]. Additionally, Abubaker et al. [36] also reported comparable NR value of 5.9 g/day for sheep fed on cottonseed cake in complete diet. Therefore, this shows that MKC may be used to replace some conventional plant protein supplements, such as sunflower seedcake or cottonseed cake.

### 4.3. Growth Performance

The ADG and final body weight of the lambs was similar across the treatments, regardless of the difference in sources of protein in the different diets. The reason for this is that the diets were formulated to be isonitrogenous and isoenergetic, hence the lambs grew in a similar fashion by having a similar ADG across treatments. This suggests that all treatments supplied comparable nutrients [37] needed for maintenance and growth by the lambs. Similarly, Yagoubi et al. [7] also reported non-significant differences in body weight gain of lambs fed different nitrogen sources (soybean meal and *Vicia faba* bean). In contrast, Ruzic-Muslic [9] reported significant differences on ADG of lambs fed on complete diets with different protein sources (sunflower meal, soybean meal, and fish meal). In the study by Ruzic-Muslic [9], both soybean meal and fish meal complete diets had superior ADG than sunflower meal diet, suggesting that a high content of RUP made the difference among the three protein sources. Fish meal is known for its high biological value as a protein of animal origin [18], while soybean meal has lower fibre but better quality of protein, as determined by essential amino acids levels [38]. Live weight at slaughter (final weight) is an important variable as it determines both HCW and cold carcass weight (CCW). According to Assan [39], nutrition plays a pivotal role in promoting ideal carcass characteristics and meat quality attributes. In the current study, HCW and EBW were similar across the treatments. The similarity of these traits (HCW and EBW) is probably due to similarities in the intake of both gross energy and protein (Table 3) and their utilisation. Therefore, the lambs had similar amounts or quality of nutrients availed for absorption and assimilation.

### 4.4. Meat Quality Attributes and Proximate Composition

The protein content of the *Longissimus dorsi* muscle in the current study ranged from 26.9% to 28.3%, while fat content varied from 4% to 6.9%, without any significant differences across diets. The protein, fat, and ash values of meat from the present study were not influenced by treatments and are slightly higher than results reported by Kotsampasi et al. [40] on the *Longissimus thoracis et lumborum* muscle of Greek breed Florina sheep fed graded levels of olive cake in complete diets. The content of moisture was similar to results of a previous study by Chai et al. [41].

The type of diet consumed by livestock affects meat quality attributes such as colour, pH, and tenderness. According to Chai et al. [41], the changes in muscle lightness (L*) and yellowness (b*) show the dietary effects of pre-slaughter glycogen and marbling levels. The results on lightness (L*), redness index (a*), and yellowness (b*) across treatments were not different and are comparable to findings of Bezerra et al. [42] on the *Longissimus lumborum* from intact male crossbred lambs (aged 5 months) fed graded levels of peanut cake in complete diets. The results (Table 7) on lightness (L*), redness index (a*), and yellowness (b*) in the current study were within the accepted range of 30.03 to 49.47, 8.24 to 23.53, and 3.38 to 11.10, respectively, for lamb meat [42]. In addition, Facciolongo et al. [43] reported no treatment effect on the colour of meat when comparing linseed with soybean in complete diets.

The pH in a muscle of a live animal is about 7.1 [4]. Immediately after slaughter, glycogen stored in animal muscles is converted to lactic acid through anaerobic glycolysis thereby reducing the pH to the range of 5.4 to 5.7 after 24 h post-mortem [4]. The ultimate pH is used to determine the quality of meat under commercial meat production. Depletion of muscular glycogen reserves before slaughter results in poor quality meat with a high ultimate pH (more than 5.7) and renders meat susceptible to immediate bacterial spoilage [44]. In the current study, the ultimate pH across treatments was not significantly different, as was observed by Chiafalo et al. [5], and was within the acceptable range of 4.8 to 5.0, assuming that an ultimate pH greater than 5.7 is classified as undesirable [4].

### 4.5. Fatty Acid Composition of Longissimus Dorsi Muscle of Lambs

Nutritional properties of meat are also determined by fatty acid composition among other nutritional attributes [45]. High intake of saturated fatty acids (SFAs) from consuming of red meat by humans is linked to cancers and coronary heart disease. However, recently research has started to demystify negative image of meat, especially that meat consumption provides humans with some essential nutrients [46], which are important for the growing youth and the elderly [47]. In the current study, palmitic acid was the predominant (27.7% to 30.7%) saturated fatty acid across the treatments (Table 8). Palmitic acid and myristic acid are considered as being hypercholesteraemic because they have a negative influence on the blood plasma low density lipoproteins (LDL) and high-density lipoproteins (HDL) content [48]. However, lipid research would suggest that not all SFAs have the same impact on serum cholesterol, such as myristic (C14: 0), which has a greater total cholesterol raising effect than palmitic acid (C16: 0) [49]. Low density lipoproteins are linked to cardiovascular risk in humans. Oleic acid is known for reducing the risk of cholesterol accumulation [50] in humans by increasing HDL-cholesterol in blood. The high oleic acid content in the *Longissimus dorsi* muscle of lambs fed MKCD is attributed to some oleic acid from the diet escaping rumen biohydrogenation and being absorbed in the small intestine for tissue assimilation [45]. When comparing several lipids with different fatty acids profiles, Fiorentini et al. [30] noted that protected fats (saponification process where fatty acids from soybean oil reacted with calcium hydroxide) decreased biohydrogenation. Stearic fatty acid is regarded as a neutral fatty acid that does not contribute to cardiovascular diseases in humans and normally ranges from 10 to 20% of fats produced by ruminants [42]. According to Bas et al. [51], polyunsaturated fatty acid (PUFA) in sheep ranges from 4.6% to 12.5%; in the current study, PUFA content ranged from 2.9% to 5.3%, with the *Longissimus dorsi* muscle of lambs fed CD having relatively higher levels of PUFA, although this was not significantly different from other treatments. This, therefore, suggests that most PUFAs were either biohydrogenated in the rumen or were not available in large quantities from the diets of the current study. Indexes, such as the stearic acid plus oleic acid to palmitic acid ratio, describe the possible beneficial effects of different lipids found in red meat with an acceptable range of 2.1 to 2.8 for sheep meat [40]. In the current study, the ratio was satisfactory at 2.4 for the *Longissimus dorsi* muscle from lambs fed MKCD, and for both CD and SCD it was unsatisfactory at 1.3 and 0.7, respectively.

### 4.6. Sensory Analysis

The findings from the current study revealed that dietary treatments did not cause differences in the sensory parameters evaluated, i.e., appearance, taste, flavour, tenderness, juiciness, and overall impression. Consequently, the panellists’ score for all the evaluated sensory traits were higher in meat from lambs fed MKCD, but this was not significantly different from other treatments. Interestingly, sensory tenderness and instrumental tenderness were better (numerically) in meat from lambs fed MKCD, which also had numerically lower pH. According to Sen et al. [52], tenderness is the most important textural attribute of meat and is the main determinant of consumer acceptance of meat. In most cases, older animals tend to have tough meat and younger animals more tender meat. This is attributed to development and increase of connective tissues in muscles of animals as they age [52]. In the current study, all lambs were contemporaries (aged 12 months at slaughter), hence, there were no differences observed in tenderness as a result of age or diet [53]. The sensory attribute of juiciness in meat steaks is determined by water retention, lipid content, flavour, and texture. In general, the sensory meat steak ratings of the cooked meat steaks were non-significant, and this could also be explained by non-significant physico-chemical parameters. Therefore, meat evaluated in the current study is placed at a rating score of “like moderately”.

### 4.7. Gross Margin Analysis

Livestock production entails feeding of animals with various feedstuffs to attain product output in a projected period of time. Feed costs from feeding oilseed cake diets accounted for 51% while commercial diet accounted for 71% of total variable costs, which excluded purchase price of lambs. Partial or complete replacement of maize grain or conventional protein sources in animal feeding may reduce pressure on conventional energy and protein feed ingredients [54]. Feeding lambs on MKCD and SCD resulted in similar gross margins while using the CD led to high variable costs. Similarly, Chingala et al. [55] reported lower gross margins from steers fed on soybean meal complete diets than in steers fattened by Baobab (*Adansonia digitata*) complete diet and white thorn tree (*Vachelia polyacantha*) leaf meal diet, respectively. The use of MKC in complete lamb diets is a viable option to adopt, especially during periods when the cost of conventional protein sources is expensive due to high demand and limited supplies.

## 5. Conclusions

Incorporation of MKC at 12% of the total diet dry matter of lamb diets supported growth performance and yielded similar results as the commercial diet or sunflower seed cake diet. In addition, MKC has the potential to improve meat sensory attributes, as evidenced by numerically higher-ranking scores on overall eating quality. Positive gross margins are dependent on live weights of sold animals and the cost of added weight to the marketed animal. Morula kernel cake diet and SCD had similar gross margins. Therefore, the use of alternative non-conventional protein sources like MKC can help reduce feed costs without compromising growth performance (ADG), carcass traits (HCW and EBW), and meat quality attributes.

## Figures and Tables

**Table 1 animals-13-01387-t001:** Ingredients used in experimental diets.

	Treatments	
Ingredients (%)	MKCD	SCD
Maize grain ^1^	59	61
Sorghum stover	24	21
Sunflower seedcake	-	10
Morula kernel cake	12	-
Wheat bran	2.4	5
Urea	0.5	1.2
Liquid molasses	1.2	1.4
Feed lime	0.5	0.7
Dicalcium phosphate	0.5	0.5
Salt	0.5	0.1

MKCD = morula kernel cake diet; SCD = sunflower seedcake diet; ^1^ Maize was processed by grinding into a meal.

**Table 2 animals-13-01387-t002:** Chemical composition (% on DM) and gross energy (MJ/kg) content of treatment diets.

	Treatment		
Chemical Composition	MKCD	SCD	CD
Dry matter	94.7	95.1	95.5
Organic matter	93.0	92.7	92.7
Crude protein	15.9	14.1	14.1
Ether extract	9.4	2.9	4.8
Ash	7.0	7.3	7.3
NDF	41.7	39.4	33.5
ADF	18.4	18.1	9.2
GE	8.3	8.8	8.4

MKCD = morula kernel cake diet, SCD = sunflower seedcake diet, CD = commercial diet, NDF = neutral detergent fibre, ADF = acid detergent fibre, GE = gross energy.

**Table 3 animals-13-01387-t003:** Nutrient intake (g/day, unless stated differently) from the digestibility experiment (dry matter basis) of lambs.

	Treatment				
Item	CD	MKCD	SCD	RMSE	*p*-Value
GE intake, MJ	16.9	15	16	2.2	0.2
DM intake	1008.0	890.6	954.0	129.7	0.2
OM intake	934.4	828.3	884.3	120.3	0.1
CP intake	142.1	141.6	134.5	18.7	0.5
EE intake	48.4 ^b^	83.7 ^a^	27.9 ^c^	7.8	0.0002
NDF intake	367.4	371.4	375.4	55	0.5
ADF intake	92.7 ^b^	163.9 ^a^	172.7 ^a^	18.7	0.004
Ash intake	73.6	62.4	69.6	3.4	0.1

CD = commercial diet, MKCD = morula kernel cake diet, SCD = sunflower seedcake diet, RMSE = root mean standard error, GE = gross energy DM = dry matter, OM = organic matter, CP = crude protein, EE = ether extract, NDF = neutral detergent fibre, ADF = acid detergent fibre, ^a,b,c^ means in a row with similar superscripts do not differ (*p* > 0.05).

**Table 4 animals-13-01387-t004:** Apparent nutrient digestibility (% dry matter basis) by the lambs.

	Treatments				
Item	CD	MKCD	SCD	RMSE	*p*-Value
Dry matter	92.7	81.3	83	4.1	0.08
Organic matter	92.9	82.3	80.6	5.1	0.07
Crude protein	92.3 ^a^	84.8 ^b^	82.5 ^b^	2.8	0.01
Ether extract	89.1	84.9	72.5	8.8	0.1
NDF	84.7	70.0	65.6	7.4	0.2
ADF	76.7	62.0	68	7.4	0.2

CD = commercial diet, MKCD = morula kernel cake diet, SCD = sunflower seedcake diet, RMSE = root mean standard error, NDF = neutral detergent fibre, ADF = acid detergent fibre, ^a,b^ means in a row with similar superscripts do not differ (*p* > 0.05).

**Table 5 animals-13-01387-t005:** Nitrogen balance in lambs fed different diets.

	Treatment				
Item	CD	MKCD	SCD	RMSE	*p*-Value
N-intake (g/d)	21.5	20.5	20.4	3.1	0.5
Excretion (g/d)					
Urinary-N	8.5	8.7	9.8	2.3	0.8
Faecal-N	1.5	2.6	3.8	0.8	0.06
Total	10.1	11.3	13.6	2.6	0.4
Absorbed-N	20.0	18.0	16.6	2.8	0.2
N-Retention					
N-Retention (g/d)	8.7 ^a^	6.4 ^ab^	3.9 ^b^	1.9	0.03
Retention-I (%)	39.9 ^a^	31.6 ^ab^	19.1 ^b^	7.7	0.03
Retention-A (%)	43.1	35.8	23.6	8.9	0.07

CD = commercial diet, MKCD = morula kernel cake diet, SCD = sunflower seedcake diet, RMSE = root mean standard error, N = nitrogen, d = day, I = intake, A = absorbed, ^a,b^ means in a row with similar superscripts do not differ (*p* > 0.05).

**Table 6 animals-13-01387-t006:** Growth performance and carcass traits of lambs given the treatments.

	Treatment				
Item	CD	MKCD	SCD	RMSE	*p*-Value
Performance					
Initial weight (kg)	16.2	16.7	17.9	0.5	0.9
Final weight (kg)	35.3	34.0	35.1	2.3	0.6
DMI (g)	890.4	867.8	905.9	71.5	0.7
ADG (g)	176.3	163.7	174.3	22.8	0.6
FCR	5.7	6.2	5.9	0.8	0.6
Carcass traits					
EBW (kg)	29.1	28.5	30.3	2.0	0.3
HCW (kg)	17.1	16.9	17.1	1.0	0.2

CD = commercial diet, MKCD = morula kernel cake diet, SCD = sunflower seedcake diet, RMSE = root mean standard error, DMI = dry matter intake, ADG = average daily gain, FCR = fed conversion ratio, EBW = empty body weight, HCW = hot carcass weight.

**Table 7 animals-13-01387-t007:** Physico-chemical attributes and proximate composition of the *Longissimus dorsi* muscle from lambs.

	Treatments				
Items	CD	MKCD	SCD	RMSE	*p*-Value
Physico-chemical					
Shear force, N	22.9	20.1	24.0	4.3	0.3
pH_24h_	5.0	4.8	5.0	0.4	0.8
L*	39.4	40.9	43.7	7.3	0.6
a*	20.7	21.6	22.3	2.4	0.6
b*	7.6	8.9	8.3	1.4	0.3
Proximate composition					
Moisture (%)	73.5	72.5	72.5	2.2	0.7
Protein (%)	28.3	27.5	26.9	3.6	0.8
Fat (%)	4.0	5.2	6.9	3.0	0.3
Ash (%)	1.2	1.3	1.2	0.3	0.8

CD = commercial diet, MKCD = morula kernel cake diet, SCD = sunflower seedcake diet, RMSE = root mean standard error, N = newton, L* = lightness, a* = redness, b* = yellowness.

**Table 8 animals-13-01387-t008:** Fatty acid profile (% of total identified fatty acid) of the *Longissimus dorsi* muscle of lambs.

	Treatment				
Item	CD	MKCD	SCD	RMSE	*p*-Value
Saturated					
14:0	2.3	2.2	2.9	0.3	0.1
16:0	27.7 ^b^	27.7 ^b^	30.7 ^a^	0.6	0.01
18:0	20.5	20.8	20.8	9.9	0.3
Monounsaturated					
18:1n-9	14.3 ^b^	46.1 ^a^	ND	15.7	0.05
18:1ME^1^	ND	ND	42.5	13.7	0.2
18:1methyl ester^2^	15.0	ND	ND	16.4	0.5
19:1cis-13	14.9	ND	ND	16.3	0.5
Polyunsaturated					
18:2n-6	1.8	ND	1.8	2.3	0.6
18:2ME^3^	ND	0.9	ND	1.0	0.5
18:2n-7	1.9	ND	ND	2.1	0.5
19:2(8E,11E)^4^	1.6	0.9	ND	2.0	0.7
∑SFA	50.5	50.9	54.3	1.3	0.06
∑MUFA	44.2 ^ab^	46.1 ^a^	42.5 ^b^	1.0	0.02
∑PUFA	5.3 ^a^	2.9 ^b^	3.2 ^b^	0.4	0.02

CD = commercial diet, MKCD = morula kernel cake, SCD = sunflower seedcake diet, RMSE = root mean standard error, ND = not detected, 18:1ME^1^ = methyl 10-Octadecenoate, 18:1methyl ester^2^ = trans-vaccenic, methyl ester,18:2ME^3^ = methyl 9-cis,11-trans-Octadecadienoate, 19:2(8E,11E)^4^ = 8,11-Octadecadienoic acid methyl ester, SFA = saturated fatty acids, MUFA = monounsaturated fatty acids, PUFA= polyunsaturated fatty acids, ^a,b^ means in a row with similar superscripts do not differ (*p* > 0.05).

**Table 9 animals-13-01387-t009:** Sensory evaluation of the *Longissimus dorsi* muscle from lambs.

	Treatment				
Item	CD	MKCD	SCD	RMSE	*p*-Value
Appearance	6.5	7.2	6.9	1.8	0.5
Taste	7.1	7.7	7.0	1.4	0.8
Flavour	7.4	7.5	7.3	1.6	0.3
Tenderness	7.2	7.7	7.0	1.6	0.3
Juiciness	6.8	7.6	7.2	1.6	0.3
Overall impression	7.5	8.0	7.3	1.0	0.09

CD = commercial diet, MKCD = morula kernel cake diet, SCD = sunflower seedcake diet, RMSE = root mean standard error, rating scale; 1 = dislike extremely to 9 = like extremely.

**Table 10 animals-13-01387-t010:** Gross margin analysis (per lamb) of the three treatment diets.

	Treatment				
Item (BWP *)	CD	MKCD	SCD	RMSE	*p*-Value
Hot carcass value	1333.20	1321.60	1431.10	140.5	0.3
Edible offal value	55.61	53.39	58.73	6.3	0.7
Head and feet value	34.80	36.40	34.80	3.0	0.6
Total output	1404.60	1389.10	1505.40	146.2	0.4

Feed	420.40	242.10	244.70	-	-
Weaner	800.00	800.00	800.00	-	-
Abattoir	98.90	98.90	98.90	-	-
Drugs	4.40	4.40	4.40	-	-
Labour	55.60	55.60	55.60	-	-
Transport	13.90	13.90	13.90	-	-
Total variable costs	1390.50 ^a^	1277.30 ^b^	1292.70 ^b^	129.7	0.0001
GM	14.00 ^b^	111.70 ^ab^	212.70 ^a^	129.7	0.05

CD = commercial diet, MKCD = morula kernel cake, SCD = sunflower seedcake diet, RMSE = root mean standard error, GM = gross margin, * 1USD=10 BWP, ^a,b^ means in a row with similar superscripts do not differ (*p* < 0.05).

## Data Availability

The data presented in this study are available on request from the corresponding author.

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
