# Peer review of "Morula Kernel Cake (Sclerocarya birrea) as a Protein Source in Diets of Finishing Tswana Lambs: Effects on Nutrient Digestibility, Growth, Meat Quality, and Gross Margin"

_animals, 2023, doi:10.3390/ani13081387_

Round 1
Reviewer 1 Report
Dear authors
I stated a deep commitment in your manuscript. I found also several vagueness and i suggest some improvements.
All the best

Author Response
Please see the attachment, thanks.
Response to Reviewer 1 Comments
Simple Summary, line 12: sup-ported?
Response: corrected in line 12
Simple Summary, line 13: I found at page 7, line 222 (Table 6) the acronym and I agree, however I suggest you to put the acronym of feed conversion ratio (FCR) in the simple summary at the first time, or in the abstract, if you prefer. Thanks.
Response: acronym used for the first time under simple summary in line 13
Simple Summary, line 16: reten-tion? (other typo ambiguities at lines 19, 26 -com-plete-, nutritionists at line 53…
Response: all corrected as advised
Abstract, lines 23-24: improve sentence please (grammar)
Response: It has been modified as follows: A trial evaluated growth performance, carcass characteristics, meat quality and economic returns of fattened lambs fed on diets containing different protein sources.
Abstract: I saw the binomial term of Morula Kernel Cake in the title, it is ok. However I suggest you to indicate Sclerocarya birrea (italic) in the abstract also.
Response: scientific name for Morula kernel cake introduced under abstract section at line 34.
Introduction, line 48-49: Improve ruminal microbes, please adopt better representation. According to Chiofalo et al. (2020) [5].
Response: line 48-49 modified under line 189……….. Therefore, it reads as ‘According to Chiofalo et al. [5] rumen microbes hydrogenate dietary unsaturated fatty acids resulting in the production of saturated fatty acids that are absorbed and and deposited in animal tissues
Introduction, line 51-58: Additionally, Junkuszew et al. (2020) [6]; as stated by Yagoubi et al. (2018) [7] and other typo errors in the manuscript (e.g. lines 74, 170, 278).
Response: changes are as follows: Junkuszew et al. [6] stated that dietitians recommend consumption of meat low in fat and rich in n-3 and n-6 fatty acids to avoid heart-related diseases in humans. Other typos were as requested.
Introduction, line 61-63: Please, improve the sentence “feeding strategies aimed at…” (grammar).
Response: modified at line 202 and read as…Therefore, provision of protein or energy supplements will promote better utilisation of poor quality roughages by ruminants.
Introduction, line 65-67: I think you can delete this phrase “Ruminants usually require quality protein…”, is redundant.
Response: phrase deleted as advised.
Introduction, line 72: MKC, ok here the acronym fits.
Materials and methods, lines 83-86; ethics statement. Add the documents (pdf) you mention, please.
Response: The paragraph modified and document on guidelines of council for the international organisation of medical sciences (CIOM, 2012) has been cited.
Materials and methods, line 92: what kind of internal parasites?
Response: round worms and tapeworms.
Materials and methods, line 97: straw litter pens? Or other materials, or concrete without straw?
Response: concrete floors.
Materials and methods, line 100: energy=78% TDN. Can you explain please?
Response: TDN stands for total digestible nutrients and diets were formulated using Aries software (California, USA).
Materials and methods, line 100: I see the word “sheep” in this chapter the first time. I found “sheep” in the title and simple summary before. Please, be careful when using the term sheep or lamb, especially in experimental design.
Response: Lamb used throughout the document.
Materials and methods, in Table 1. What is ground maize grain? Moreover I suggest to adopt corn rather than maize (in the entire manuscript).
Response: ground maize refers to maize grain that was processed into a meal by grinding. Under Table 1: a foot note is typewritten to explain. Since the whole document was written using British English instead of American English, therefore maize will not be changed to corn (which is used when writing using American English).
Materials and methods, line 117: As still suggested, lamb or sheep!?
Response: lamb is now used in throughout the whole document as mentioned earlier.
Materials and methods, lines 132-133: Please check bibliography (Ekeocha et el. 2012) and formula, moreover others literature sources are required.
Response: Under Bibliography Ekeocha included as reference number 19 in the list. Regarding the formula another reference used is McDonald et al. 2011.
Materials and methods, lines 138: I’m sure, animals were humanely slaughtered. Please describe the procedure (stunning, slaughtering, jugulation, bleeding). Thanks.
Response: At line 407 inserted “The lambs were electrically stunned and immediately bled by cutting the jagular vein and carotid arteries using a sharp knife.
Materials and methods, Statistical Analysis: statistical analysis is not described, there is only the formula, moreover GLM (general linear model) acronym requires a previous description. The bibliographic reference to SAS software is not sufficient.
Response: General liner model described as advised at line 492. At line 495 to 497inserted “The significance of the treatment effects were verified using the F-test and for all measured parameters, means were separated Tukey’s procedure, level of significance was declared at p < 0.05 while tendencies was declared at p < 0.1.
Materials and methods, in Table 3. I suggest to indicate GE intake (as MJ/day) separately from other ingestions, the unit of measurement is different (g/day). You can put GE simply at the first line for example.
Response: GE put first in the table as advised. Further to that the title of table was modified to read “Nutrient intake (g/day, unless stated differently) from digestibility experiment (dry matter basis) of lambs. Words in the brackets are the ones modified.
Results, lines 199-200. Can you test again significance of dry matter and organic matter digestibility? you will probably reconsider the tendency (p=0.07 and p=0.08 respectively), stating no significant results.
Thanks. Similarly for Retention-A in nitrogen balance (p=0.07). Ok tendency of ∑SFA (p=0.06), I agree. Test
Response: tendency was declared at p < 0.1, I erred by not stating that under statistical analysis.
again 18:1n-9, too close to the threshold value. Where is the table of gross margin analysis? Why in discussion?
Response: Table 10 (gross margin) placed under results section as expected.
Discussion, lines 266. Improve the sentence as follows: Ahmed et al. (2003) [28] reported non-significant dry matter intake (DMI)…
Response: sentence adopted as suggested.
Discussion, lines 287. I suggest to change with: Different intake of nitrogen in the present study did not show statistically significant differences.
Response: sentence adopted as suggested.
Discussion, lines 292-293, 296: please adopt the word lamb (versus sheep)!
Response: adopted as suggested.
Discussion, line 306: Vicia faba bean, please.
Response: corrected
Discussion, lines 333, 335-336: Please improve the sentence “The lack of effect of diet on the colour of meat” (grammar), and “An animal has a certain amount of energy stored in the muscle as glycogen” (unfit with a scientific article).
Response: modified at line 1101 to 1103 and reads “Immediately after slaughter, glycogen stored in animal muscles is converted to lactic acid through anaerobic glycolysis thereby reducing pH to the range of 5.4 to 5.7 after 24 h post-mortem [4]”.
Conclusions, lines 415-418. I would suggest greater caution “reduced feed costs without any detrimental effect on the…”
Response: line 1184 to 1187 modified to “Therefore, the use of alternative non-conventional protein sources like moula kernel cake ( MKC) can help reduce feed costs without compromising growth performance (ADG), carcass traits (HCW and EBW) and meat quality attributes.”
Reviewer 2 Report
Page 1, line 12: supported
Page 1, line 14: Lucerne
Page 1, line 16: retention
Page 1, line 16: leave only MKCD
Page 1, line 19: Furthermore
Page 2, line 44: demanding
Page 2, line 46: in human diets
Page 2, line 53: nutritionists
Page 2, line 72: provide full name of MKC when first mentioned
Page 2, line 81: why did you choose sunflower seed cake?
Page 2, line 88: 8 months old? Please check it
Page 3, line 101: you reported sheep were 7 months old, while in line 88 you reported that sheep were 8 months old. Please, clarify this.
Page 3, Table 1: Information about control diet composition is lacking. Please, provide data on ingredients used for control diet (CD)
Page 3, line 103-104: provide information about the diet offered to lambs during adaptation period? Did you include fasting period?
Page 4, line 149: provide the country of the producer
Page 5, line 179: provide information about the number of replications
Page 5, line 181: provide full name of CRD. Generally, information about statistical analysis is incomplete, it should be improved.
Page 7, line 235: MUFAs were not significantly higher in CD of lambs fed MKCD compared to that of lambs fed CD, while was higher compared to that of lambs fed SCD.
Page 9, line 276: delete latin name of MKC
Page 10, line 318: Use either Longissimus dorsi or LD. Please, uniform the it throughout the manuscript.
Page 11, line 358-359: support the statement with an appropriate reference
Page 11, line 369-370: provide the information about the mentioned index in Table 8
Page 11, line 375: replace being with i.e.
Page 11, section 4.7: increase the font
Authors should check the typing throughout the manuscript and citation of references in the text. For instance, in line 58: reference is cited „as stated by (7)...“ , but should be Yagoubi et al. (7). This stands also for other references.
Author Response
Good day, please see the attachment, thanks.
Page 1, line 12: supported
Response: corrected
Page 1, line 14: Lucerne
Response: corrected
Page 1, line 16: retention
Response: corrected
Page 1, line 16: leave only MKCD
Response: not changed as advised because this is where it was first defined.
Page 1, line 19: Furthermore
Response: corrected
Page 2, line 44: demanding
Response: corrected
Page 2, line 46: in human diets
Response: corrected as suggested here…in human diets
Page 2, line 53: nutritionists
Response: corrected
Page 2, line 72: provide full name of MKC when first mentioned
Response: defined under summary at line 11
Page 2, line 81: why did you choose sunflower seed cake?
Response: it is a common conventional plant protein source grown and used in livestock diets in Botswana. Therefore, in the current study MKC which is potential feed ingredient was compared to sunflower seedcake as example of available conventional protein sources.
Page 2, line 88: 8 months old? Please check it
Response: corrected accordingly.
Page 3, line 101: you reported sheep were 7 months old, while in line 88 you reported that sheep were 8 months old. Please, clarify this.
Response: adjusted as appropriate in line 289.
Page 3, Table 1: Information about control diet composition is lacking. Please, provide data on ingredients used for control diet (CD)
Response: only ingredients given by the manufacturer (stated in line 291 to 293) were provided as the formula is confidential.
Page 3, line 103-104: provide information about the diet offered to lambs during adaptation period? Did you include fasting period?
Response: Details provided as an improvement in line 294 to 296: Allocation of the experimental diets during the adaptation period was restricted to avoid digestive disorders. After the adaptation period, each lamb was fed ad libitum on a daily basis with 15% allowance of leftovers in their individual pens for 103 days.
Page 4, line 149: provide the country of the producer
Response: provided as advised (New York, USA) in line 422.
Page 5, line 179: provide information about the number of replications
Response: replications are provided at respective sections; line 282, line 303, line 417 and line 461.
Page 5, line 181: provide full name of CRD. Generally, information about statistical analysis is incomplete, it should be improved.
Response: CRD now inserted at line 283 and defined as suggested. Information on statistical analysis improved as advised. Inserted words at line 501….” general linear model (GLM) procedure of SAS’ and line 504 to 506 modified to ‘The significance of the treatment effects were verified using F-test and means were separated using Tukey’s procedure. Level of significance was declared at p < 0.05 and for tendencies significance was declared at p < 0.1.
Page 7, line 235: MUFAs were not significantly higher in CD of lambs fed MKCD compared to that of lambs fed CD, while was higher compared to that of lambs fed SCD.
Response: the sentence was slightly modified as shown and throughout the document LD is no longer abbreviated. “Monounsaturated fatty acids were higher in Longissimus dorsi muscle from lambs fed on MKCD and lower in Longissimus dorsi muscle from lambs fed SCD” (from line 747 to 748).
Page 9, line 276: delete latin name of MKC
Response: deleted as advised.
Page 10, line 318: Use either Longissimus dorsi or LD. Please, uniform the it throughout the manuscript.
Response: Longissimus dorsi used throughout the document instead of LD as advised.
Page 11, line 358-359: support the statement with an appropriate reference
Response: reference provided at line 1137.
Page 11, line 369-370: provide the information about the mentioned index in Table 8
Response: the information about the index on other treatments is provided in-text and I thought adding another row will make the table too big, hence going for the former. Information is provided at line 1149 to 1151.
Page 11, line 375: replace being with i.e.
Response: changed as advised.
Page 11, section 4.7: increase the font
Response: font adjusted accordingly.
Authors should check the typing throughout the manuscript and citation of references in the text. For instance, in line 58: reference is cited „as stated by (7)...“ , but should be Yagoubi et al. (7). This stands also for other references.
Response: concern raised addressed accordingly.

Reviewer 3 Report
This study by Baleseng et al. evaluated growth performance, carcass characteristics, meat quality and gross margins of fattened lambs fed complete diets containing different protein sources. The study is focused, well-designed and experimental procedures are clearly described.
The manuscript is clearly written, the discussion is conducted correctly, the conclusions are concise and reflect the results well, but a little short. I think the article is almost ready for publication, however I have some comments, please find it below.
Comments
Simple Summary and Abstract:
line 12, 14, 19, 26 and more – in these places I found words with '-' inside, please correct it in these paragraphs and in the whole article
Methods
Morula is not a plant normally used in animal nutrition, so I failed to indicate its chemical composition and compare it with the chemical composition of sunflower.
Results
Table 7 - the headers are a bit different
References
Please correct the numbering in the references.
Point 2 and 31 - there are an unnecessary duplicate space here
Author Response
Good day, please see the attached, regards.
Response to Reviewer 3 Comments
Simple Summary and Abstract:
line 12, 14, 19, 26 and more – in these places I found words with '-' inside, please correct it in these paragraphs and in the whole article
Response: The corrections were done as advised throughout the document.
Methods
Morula is not a plant normally used in animal nutrition, so I failed to indicate its chemical composition and compare it with the chemical composition of sunflower.
Response: In line 213 Malebana et al. [11] is cited as they have fully profiled chemical composition of MKC.
Results
Table 7 - the headers are a bit different
Response: At footnote of Table 7 MKD was corrected to MKCD as expected.
References
Please correct the numbering in the references.
Response: all references were corrected and numbering adjusted as required.
Point 2 and 31 - there are an unnecessary duplicate space here
Response: space at reference 2 corrected and for previous reference 31 which is now 32, space has been corrected also.
Reviewer 4 Report
Thank you for the article sent, however, I have a few comments on it:
-line 12, the results did not show a statistically significant effect of the MKC on the traits listed,
- lines 32- 34, if the MKC diet is taken as an experimental factor, then I think it should be mentioned in the results first, but if the SC and MKC diets are taken as such a factor (line 99), then I think the title of the article should mention both diets,
- what was the lambs' diet like before they started getting the finishing diet? For an 8 month old, compared to many breeds, their body weight is relatively low
- Table 1. information about the composition of the control diet is missing
- line 111, why only 4 lambs from each group?
- for the tables with the results, there is no information about the standard deviation for the tested values, the averages themselves indicate greater differences, for example, in Table 6. or 10.
- line 193, the "C" in the abbreviation "MKCD" is missing.
- line 223, from where the abbreviation "Wt=weight"
- no information about the fatty acid composition of individual diets
- table 8., why for the SC diet the sum of acids so significantly deviates from 100%?
- table 10. should either be in the results, or it should be mentioned in which table the results for gross margin analysis are located
- lines 412- 413, no statistical differences
- please note the additional "." and "," in the references
Author Response
Good day, please see attachment, thanks for your feedback.
Response to Reviewer 4 Comments
line 12, the results did not show a statistically significant effect of the MKC on the traits listed,
Response: True that, however, in the current study, Morula kernel cake (MKC) as a potential feed ingredient that was compared with the conventional protein sources being sunflower seedcake and Lucerne which was used in the commercial diet (Sheep finisher being sold in the local market). With regard to growth the emphasis being made here was that MKC was as good as sunflower seedcake or Lucerne in the current study as a source of nitrogen in the diet. For sensory attributes in terms of overall impression (P=0.09) and with regards to trends in which significance was declared at P < 0.1 the statement is justified. I must also point out that at the time you gave your response, my statistical methods section in the manuscript lacked the statement I provided in the previous sentence (so your critique is understandable).
- lines 32- 34, if the MKC diet is taken as an experimental factor, then I think it should be mentioned in the results first, but if the SC and MKC diets are taken as such a factor (line 99), then I think the title of the article should mention both diets,
Response: clarity is stated as above. I think the title is ok as is and it can also be written as you suggest. In all the tables presented, I started all the results (in Tables) with the commercial diet (CD) which was used as the control, followed by MKCD in which the potential feed ingredient (MKC) was used. So in most publications that I have seen the norm is usually to start results presentation (in tables) with the control and then proceed with your other treatments of interest. I’m also open to you suggesting a potential topic for this manuscript after the clarifications that I have just put forward. (I hope I have not misread your comments or suggestions).
- what was the lambs' diet like before they started getting the finishing diet? For an 8 month old, compared to many breeds, their body weight is relatively low
Response: In line 294 to 297 the write-up was improved as such “Allocation of the experimental diets during the adaptation period was restricted to avoid digestive disorders. After the adaptation period, each lamb was fed ad libitum on a daily basis with 15% allowance of leftovers in their individual pens for 103 days’. Yes, the Tswana sheep is indigenous to Botswana and can be classified as a small-sized sheep in terms of body weight when compared to other breeds like Dorper or other breeds found in countries like Brazil or USA.
- Table 1. information about the composition of the control diet is missing
Response: composition of the commercial diet is not available as it is a business secret. But the ingredients for the commercial diet were yellow maize, maize bran, feed lime, ammonium chloride, liquid molasses, salt and vitamin premix (feed dealer; personal communication). This information was inserted at line 291 to 293 to address the concern you raised.
- line 111, why only 4 lambs from each group?
Response: when doing metabolism studies like in the current study, the minimum number required is three animals per treatment (McDonald et al. [18]). So in our case we used four lambs per treatment to improve the quality of our data and the resources (total number of metabolism crates) we had allowed us to use such replications per treatment.
- for the tables with the results, there is no information about the standard deviation for the tested values, the averages themselves indicate greater differences, for example, in Table 6. or 10.
Response: Thanks for your observation, what you are suggesting is another option of presenting the results (presentation of standard deviation), In the present study, the way we presented the results in the tables is okay especially that treatment means for a given parameter were presented with their respective root mean standard error (which is equivalent to standard error of the mean) of the treatment means. The standard error of the mean was also within the acceptable range for all parameters measured. However, regarding sample analysis in the laboratory, it was done in duplicates for a given parameter and if the standard deviation or coefficient of variation was too high laboratory analysis was repeated.
- line 193, the "C" in the abbreviation "MKCD" is missing.
Response: footnote in Table 3 corrected to MKCD as advised.
- line 223, from where the abbreviation "Wt=weight"
Response: WT has been deleted.
- no information about the fatty acid composition of individual diets
Response: Thanks for the observation, due to budgetary constraints fatty acids in the experimental diets were not profiled.
- table 8., why for the SC diet the sum of acids so significantly deviates from 100%?
Response: There was a typing error and it has been entered correctly (42.5%).
- table 10. should either be in the results, or it should be mentioned in which table the results for gross margin analysis are located
Response: The table was misplaced and a correction has been made and it is now placed in the results section as expected.
- lines 412- 413, no statistical differences
Response: The statement was improved as follows (line 1188 to 1191) … Incorporation of MKC at 12% of the total diet dry matter of lambs diets supported growth performance and yielded similar results as the commercial diet or sunflower seedcake diet. In addition, MKC has the potential to improve meat sensory attributes as evidenced by numerically higher-ranking scores on overall eating quality.
- please note the additional "." and "," in the references
Response: went through all the references to look for and correct any errors made.
-Improvement of results presentation, research design and methods
Response: the above were improved in the revised document as suggested.
Round 2
Reviewer 1 Report
I found your manuscript greatly improved. I appreciate it
Thanks
Author Response
Good day, Please see the attached, thanks.

Reviewer 2 Report
The revised manuscript has been significantly corrected and improved according to the reviewer suggestions.
Author Response
Good day, thanks for the feedback.
Reviewer 4 Report
Thank you for your responses and all the changes you have made. Personally, however, I think that the biggest deficiency in the work is the lack of fatty acid contribution to individual diets. Also referring to the 2nd comment from the previous review, I was not talking about the layout in the tables, indeed it is correct, but rather about describing the results in the text, that is, actually starting with the experimental diet, which in the case of your study is MKCD.
However, disregarding this fact, I have a few comments:
-line 8. shouldn't "Resources", start with a capital letter? (Same thing in line 86.),
-line 14., You can add "as a source of protein" at the end of the sentence,
-It seems unnecessary to me to give the number of days in brackets,
-table 3. no development of the abbreviation GE,
-in the description of the results there is no mention of statistical trend in overall impression,
-table 10. lacks a p-value.
Author Response
Good day, please see the following, thanks.
Response to Reviewer 4 Comments
Research design
Response: With regard to feeding trial, I think now it is ok but for sensory evaluation protocol, it was improved by inserting a sentence at line 462, “Thereafter cooked meat samples were placed in coded plates after cooling for individual sensory evaluation. After evaluation of meat steaks from each treatment the panellists were requested to sip water to neutralise their senses”.
Line 8
Response: typed ‘Resources with capital R’ and also at line 249.
lines 12:
response: inserted “as a source of protein” as you suggested.
It seems unnecessary to me to give the number of days in brackets
Response: all numbers in brackets were deleted as suggested, at line 52 and line 38
Results presentation
Response: Table 3, foot note (line 512) inserted “GE= gross energy”
Table 7, footnote MKC modified to MKCD
Line 658: inserted “However, cooked meat steaks from lambs that consumed MKCD had a tendency (p=0.09) to rank high on overall impression”.
Table 10, improved by inserting P-values and inserting all variable costs in the table.
Discussion
Response: as a result of modifying Table 10, line 984 was modified to read “The cost of feeding treatment diets ranged from 58% (oilseed cake diets) to 71% (commercial diet).